# Individualized Mini-Panel Sequencing of ctDNA Allows Tumor Monitoring in Complex Karyotype Sarcomas

**DOI:** 10.3390/ijms231810215

**Published:** 2022-09-06

**Authors:** David Braig, Alexander Runkel, Anja E. Eisenhardt, Adrian Schmid, Johannes Zeller, Thomas Pauli, Ute Lausch, Julius Wehrle, Peter Bronsert, Matthias Jung, Jurij Kiefer, Melanie Boerries, Steffen U. Eisenhardt

**Affiliations:** 1Department of Plastic and Hand Surgery, Medical Center-University of Freiburg, Faculty of Medicine, University of Freiburg, 79106 Freiburg, Germany; 2Division of Hand, Plastic and Aesthetic Surgery, University Hospital, Ludwig Maximilian University of Munich, 80336 Munich, Germany; 3Institute of Medical Bioinformatics and Systems Medicine, Medical Center-University of Freiburg, Faculty of Medicine, University of Freiburg, 79106 Freiburg, Germany; 4Department of Medicine I, Medical Center-University of Freiburg, Faculty of Medicine, University of Freiburg, 79106 Freiburg, Germany; 5Institute for Surgical Pathology, Medical Center-University of Freiburg, Faculty of Medicine, University of Freiburg, 79106 Freiburg, Germany; 6Tumorbank Comprehensive Cancer Center Freiburg, Medical Center-University of Freiburg, Faculty of Medicine, University of Freiburg, 79106 Freiburg, Germany; 7Department of Diagnostic and Interventional Radiology, Medical Center-University of Freiburg, Faculty of Medicine, University of Freiburg, 79106 Freiburg, Germany

**Keywords:** soft tissue sarcoma, complex karyotype sarcoma, ctDNA, cfDNA, biomarker, liquid biopsy, myxofibrosarcoma, leiomyosarcoma, pleomorphic sarcoma

## Abstract

Soft tissue sarcomas (STS) are rare tumors of mesenchymal origin with high mortality. After curative resection, about one third of patients suffer from distant metastases. Tumor follow-up only covers a portion of recurrences and is associated with high cost and radiation burden. For metastasized STS, only limited inferences can be drawn from imaging data regarding therapy response. To date there are no established and evidence-based diagnostic biomarkers for STS due to their rarity and diversity. In a proof-of-concept study, circulating tumor DNA (ctDNA) was quantified in (*n* = 25) plasma samples obtained from (*n* = 3) patients with complex karyotype STS collected over three years. Genotyping of tumor tissue was performed by exome sequencing. Patient-individual mini-panels for targeted next-generation sequencing were designed encompassing up to 30 mutated regions of interest. Circulating free DNA (cfDNA) was purified from plasma and ctDNA quantified therein. ctDNA values were correlated with clinical parameters. ctDNA concentrations correlated with the tumor burden. In case of full remission, no ctDNA was detectable. Patients with a recurrence at a later stage showed low levels of ctDNA during clinical remission, indicating minimal residual disease. In active disease (primary tumor or metastatic disease), ctDNA was highly elevated. We observed direct response to treatment, with a ctDNA decline after tumor resections, radiotherapy, and chemotherapy. Quantification of ctDNA allows for the early detection of recurrence or metastases and can be used to monitor treatment response in STS. Therapeutic decisions can be made earlier, such as the continuation of a targeted adjuvant therapy or the implementation of extended imaging to detect recurrences. In metastatic disease, therapy can be adjusted promptly in case of no response. These advantages may lead to a survival benefit for patients in the future.

## 1. Introduction

Soft tissue sarcomas (STS) are rare malignant tumors of mesenchymal origin with high disease-associated mortality. In localized disease, surgical resection combined with a multimodality approach allows curative therapy. However, about one third of STS patients will develop distant metastases [1]. Since prompt treatment of a recurrence improves the prognosis, a close clinical and radiological tumor follow-up has to be performed over many years for detection [2]. In addition to the high radiation exposure of >10 mSv/year during repeated computed tomography (CT) examinations and the associated risk of secondary malignancies, treatment costs are also high [3]. Furthermore, routine imaging only detects up to 75% of recurrences due to the anatomical limitations relating to the surgical site and the lung [1].

A similar situation is present in the therapeutic monitoring of disseminated STS under multimodality therapy. Determination of response with current imaging is often difficult to assess, as tumor size alone is not indicative of tumor viability. Magnetic resonance imaging (MRI) using RECIST (Response Evaluation Criteria in Solid Tumors) criteria is the gold standard assessment at most centers but has shown limited validity [4]. Therefore, new diagnostic biomarkers are urgently needed to identify patients with recurrence and predict the response of metastatic sarcomas to multimodal treatment.

Among blood-based detection methods (liquid biopsy), circulating tumor DNA (ctDNA) has been increasingly favored as a detection method in non-invasive tumor diagnostics in recent years [5]. Since tumor cells accumulate various genetic modifications during their evolution, they can be distinguished from healthy cells on this basis. Through apoptosis and necrosis of cells, free, fragmented DNA enters the peripheral circulation. This DNA is called circulating free DNA (cfDNA). A fraction of it originates from tumor cells and carries tumor-specific genetic and epigenetic modifications (ctDNA). These can be detected by highly sensitive molecular genetic methods, which can provide direct tumor detection. As a diagnostic biomarker, ctDNA can even detect minimal residual disease (MRD) in many epithelial tumors and identify patients at high risk of recurrence [6,7].

From a molecular point of view, STS can be dichotomized into STS with defined genetic alterations (most commonly translocations) and STS with various genetic alterations [8,9]. Translocation-associated STS (Myxoid Liposarcomas (MLS), Synovial Sarcomas (SS), Ewing Sarcomas (EWS), and others) account for approximately 25% of all sarcomas. They are driven by a balanced chromosomal translocation and harbor few somatic passenger mutations that commonly accumulate randomly over time. MLS, SS, and EWS have a mutational burden of only 1.7 mutations/Mb [10]. In regard to their mutational burden, MLS, SS, and EWS are thus located at the low end of the pan-cancer spectrum, in which the median tumor mutational burden ranges widely from 0.8 mutations/Mb in bone marrow myelodysplastic syndrome to 45.2 mutations/Mb in skin squamous cell carcinoma [11]. Translocation-associated STS are suitable for subtype specific ctDNA detection, as the translocation breakpoints are highly specific for each STS entity. Furthermore, each patient has an exactly defined individual translocation breakpoint within the introns [12,13]. Thus, sensitive as well as highly specific detection methods for ctDNA can be developed [14,15,16]. We have implemented and validated a targeted next-generation sequencing-based assay that combines genetic information from the breakpoint sequences with mutations identified by exome sequencing. These are integrated in patient-specific mini-panels to quantify ctDNA with high sensitivity and specificity. Tumor activity in STS can thus be monitored non-invasively with high accuracy [16,17]. Validation of the procedure in routine clinical practice is still pending.

In contrast to STS with a simple karyotype, sarcomas with a complex karyotype (Undifferentiated Pleomorphic Sarcomas (UPS), Myxofibrosarcomas (MFS), Leiomyosarcomas (LMS), and others) exhibit multiple variable genomic alterations for which a liquid biopsy approach based on recurrent hotspot mutations or translocation is not practical. They harbor large numbers of somatic copy-number alterations and a higher mutational burden [18]. Accordingly, much broader mutational profiling of the tumor is necessary to detect mutations for ctDNA quantification [19]. Due to the large genetic variability of the individual subtypes and the rarity of STS, there are, as yet, no methods for use in routine clinical practice. Here, we adapt our methodical approach based on mutational profiling by tumor exome sequencing and ctDNA quantification by tumor-individual targeted next-generation sequencing (NGS) approaches to sarcomas with complex karyotypes.

## 2. Results

Quantification of ctDNA in localized disease was performed with a 62-year-old man, who presented with a soft tissue mass of the left thigh in our department (Figure 1A). The core biopsy revealed a myxofibrosarcoma, which was poorly differentiated/high grade. Neoadjuvant radiotherapy was initiated before tumor resection. Exome Sequencing of tumor tissue obtained from the resection specimen was performed, and an individual mini-panel covering 30 mutations was generated on this basis (Figure 1B).

cfDNA was isolated from five plasma samples obtained during the course of treatment. ctDNA was enriched by hybridization capture followed by NGS sequencing. After building unique molecular identifier (UMI) consensus families, we observed a mean coverage of 1720× with a standard deviation (SD) of 347×. The mean conversion rate of the 30 regions of interest, which describes the relative number of input DNA molecules successfully converted into sequencing libraries, sequenced, mapped, and called after building UMI families, was 51.4% (SD 10.2%) (Figure 1C). Absolute quantification of ctDNA revealed 75 reads/mL on the day of biopsy (sample 1), which further increased in the following weeks before radiotherapy was initiated (sample 2). The third sample taken after completion of radiotherapy showed markedly decreased ctDNA levels (3 reads/mL) which remained constant 2 days after the surgery (sample 4). Final histopathological assessment of the tumor after resection revealed a tumor necrosis rate of 40%. There was no ctDNA detectable 3 weeks after tumor removal (sample 5). The tumor volume, as deduced from MRI imaging, decreased from 141 cm^3^ to 92 cm^3^ during the course of radiotherapy (Figure 1D).

Analyzing cfDNA, levels remained fairly constant during radiotherapy at 3–5 ng/mL (sample 1–3), increased shortly after surgery (28.4 ng/mL, sample 4), and again decreased at 3 weeks postoperatively (13.3 ng/mL, sample 5), mapping the increased cfDNA release due to surgical tissue trauma. In contrast, ctDNA, which comprised only a fraction of 0–1.43% of cfDNA, mirrored the amount of viable tumor tissue (Figure 1E).

The potential of ctDNA for surveillance of tumor recurrence was evaluated with a 42-year-old woman, who presented with a leiomyosarcoma of the lower leg. The tumor was completely resected after neoadjuvant radiotherapy, and staging revealed no signs of distant metastases (Figure 2A). Again, a patient-individual mini-panel was designed after tumor exome sequencing, which was used for targeted sequencing of 13 plasma samples collected during the course of treatment (Figure 2A). Routine follow-up according to ESMO guidelines was conducted. A CT of the chest obtained 13 months after resection of the primary tumor showed multiple new lung lesions. ctDNA, which reached baseline after resection of the primary tumor (sample 3), showed repeated low-level evidence of mutated circulating DNA fragments thereafter, steadily rising in numbers after detection of disseminated disease (sample 4–8). This could already indicate minimal residual disease (MRD) that would not be detectable by standard imaging.

The patient received three cycles of Doxorubicin/Dacarbazine; however, there was evidence of progressive disease with growing lung metastases and a new local recurrence in the lower leg. Several lung lesions, amenable to resection, were removed and a debulking operation of the leg recurrence was conducted. ctDNA increased during progressive disease (sample 9 + 10) and dropped to baseline after the two operations (sample 11 and 12), thus mirroring the amount of viable tumor tissue (Figure 2A).

Analyzing cfDNA showed low undulating concentrations around 5 ng/mL, peaking at 10–15 ng/mL after resection of lung lesions (sample 8) and the leg recurrence (sample 11) (Figure 2B). Relative amounts of ctDNA mirrored the course of the disease, and comprised 0–0.34% of cfDNA. We observed a mean coverage of 1136× (SD 207×) after UMI consensus calling and a mean conversion rate of 34% (SD 6.2%) (Figure 2C).

Monitoring of treatment response was further evaluated in a patient with a metastatic Undifferentiated Pleomorphic Sarcoma. He was included in the study one year after resection of his primary leg tumor. During follow-up, bilateral lung metastases were detected and resected in two consecutive operations, followed by adjuvant chemotherapy (Figure 3A). Exome sequencing was performed on one of the lung metastases and a mini-panel encompassing 26 tumor-associated mutations was designed. ctDNA was quantified in seven plasma samples collected during three years of treatment. ctDNA levels rose from 14 reads/mL (sample 1) to 85 reads/mL (sample 2) just before the lung lesions were resected. ctDNA declined during chemotherapy and nearly reached baseline thereafter (samples 3 + 4). Two years after the metastasectomy, a new lung lesion (0.7 cm^3^) was detected, which was subsequently removed and histologically diagnosed as a metastasis. This metastasis again caused a significant increase in ctDNA levels (sample 5: 16 reads/mL and sample 6: 22 reads/mL), followed by a decline after resection (sample 7: 3 reads/mL) (Figure 3A).

cfDNA spiked during chemotherapy (sample 3) and after resection of the 2nd lung metastasis. ctDNA values were between 0.02 and 0.43% of total cfDNA (Figure 3B). Mean coverage was 1501× (SD 202×) and conversion rate was 49% (SD 6.7%) (Figure 3C).

## 3. Discussion

In localized STS, surgical resection in a multimodal therapy concept offers a therapeutically curative approach. However, approximately thirty to fifty percent of all STS patients experience local or distant recurrence after removal of the primary tumor [1]. Local recurrences are usually detected by MRI, pulmonary metastases with conventional chest radiographs, or CT. These modalities might miss metastases at other anatomical sites, are associated with a high cost and radiation burden, and can be performed with high quality only by specialized centers. Liquid biopsy therefore offers a promising approach to complement imaging during follow-up and monitor treatment response in metastatic STS. In contrast to various mRNA-based methods, ctDNA holds great potential due to its inherent chemical stability and direct origin from genetically altered tumor cells [5]. Additionally, it allows for an accurate snapshot of the sarcoma’s genomic landscape due to its short half-life of approximately two hours [20].

Previously, we and others have presented methodologically and clinically feasible approaches for ctDNA monitoring in translocation-associated STS based on detection of patient-individual breakpoints [14,15]. This approach was adopted for routine diagnostics by implementing a disease-specific hybrid capture NGS technique in myxoid liposarcoma (MLS). By addition of tumor-derived mutations from exome sequencing, the sensitivity of this approach could be dramatically increased [16]. In this study, we adapted this diagnostic assay to the genetic landscape of STS with complex karyotypes. We could show in this proof-of-concept study that the assay enables detection of tumor recurrence and monitoring of treatment response in patients with STS, independent of the histological subtype.

Comparison of cfDNA and ctDNA measurements shows that cfDNA quantification alone does not allow for monitoring of tumor activity. In contrast to ctDNA, cfDNA can be falsely elevated by different preanalytical conditions and during the weeks following surgery [14]. This is clearly evident in patient 1, where cfDNA immediately increased sharply postoperatively, even though the tumor was completely resected. ctDNA, on the other hand, clearly reflects the proportion of vital tumor cells during treatment. The situation in patient 3 is another example, where cfDNA increased sharply during cell death due to chemotherapy, although tumor activity was very low at this time. cfDNA thus mainly indicates tissue trauma and cell death, regardless of whether the DNA releasing cells are tumor cells or healthy cells. ctDNA, on the other hand, allows for monitoring of tumor activity independent of preanalytical changes or unspecific cell death due to surgery, chemotherapy, or radiotherapy.

If used in conjunction with standard imaging during follow-up, it can improve the detection of recurrence as it monitors tumor burden throughout the body and is not restricted to the sites of imaging. This is especially important in sarcomas with an extrapulmonary spread, which occurs in one out of four patients [1]. Additionally, ctDNA levels might rise before the recurrence can be detected by imaging and therefore serve as a prognostic biomarker to distinguish low- and high-risk patients [14,16]. High-risk patients might profit from adjuvant treatment or an increased frequency of clinical follow-ups and imaging.

Monitoring treatment response is another promising area for the use of ctDNA diagnostics. Patients under neoadjuvant radiotherapy usually show a decline in ctDNA concentrations (Figure 1) [16]. A lack of decline during radiotherapy might indicate an absent response to treatment. These patients might benefit from a timely operation, rather than continued neoadjuvant radiotherapy. However, these observations and postulations have to be investigated in a prospective clinical trial. Patients who already suffer from metastatic disease can also profit from repeated ctDNA measurements to monitor (sustained) treatment response after resections and CTX. A decline in ctDNA indicates lower tumor burden, whereas an increase might prompt the treating physician to change the CTX regimen or search for further recurrences with extended imaging.

Although this approach requires tumor exome sequencing and the creation of a patient-individual target-panel, it can be largely automated in a routine diagnostic laboratory. We therefore created an analysis pipeline that evaluated tumor-specific mutations based on a matched normal sample, followed by in silico pathogenicity analysis and generation of a patient-individual panel with up to 30 target regions. Although STS with complex genotypes harbor many more mutations, previous analyses have shown that a number greater than 30 target regions only marginally improves sensitivity compared to increased sequencing costs and a similar rate of false positive results [16].

Distinguishing minimal residual disease and ctDNA levels of less than 0.01% from false positive events is a major challenge in liquid biopsy. Technically, we approach this challenge by incorporating UMIs to both ends of the library fragments and additionally taking the information from both strands during paired-end sequencing for digital error correction [16]. Evaluation of false positive results could be further improved by incorporating a leukocyte-only sample into the workflow and filtering alterations from matched white blood cells [21]. However, we have not yet implemented this step in our workflow.

Sensitivity can be improved by increasing the number of target mutations, the amount of plasma analyzed, and improvements in the workflow [22]. We incorporate up to 30 targets into our mini-panels, balancing costs for panel synthesis and sequencing and the limit of detection needed to detect small tumors and potentially even MRD. However, identifying sufficient numbers of target mutations often presents a challenge due to the low mutational burden in many STS. Even exome sequencing can sometimes detect only fewer than 30 reliable tumor mutations that can subsequently be quantified in cfDNA [18]. Besides the low mutational burden, low tumor content in the specimen, e.g., after radiotherapy, or challenging FFPE samples make mutation calling difficult. This is a particular challenge in translocation-associated STS due to the low number of passenger mutations [23].

In terms of blood volume, 5 mL of plasma is usually sufficient and can be obtained from a standard 9-mL K_2_EDTA blood collection tube. This volume can be processed with standard cfDNA extraction kits and usually yields cfDNA in a range of 10–50 ng total (2–10 ng/mL plasma) [24]. As our workflow utilizes 10 ng of cfDNA input for initial library preparation, this can be accomplished even with low cfDNA plasma concentrations.

Workflow optimization for challenging cfDNA samples also needs to be considered. The conversion rate indicates the relative number of cfDNA molecules that are successfully converted into the library, sequenced, and analyzed. We generally observe a conversion rate between 20–60%, which is very satisfactory considering the low input concentrations and the tiny panel sizes only consisting of around 5000 bases. This is accomplished by library preparations specifically adapted to cfDNA and a double hybrid-capture approach [25]. This conversion efficiency is comparable to other studies examining ctDNA in various cancers by targeted NGS approaches [26,27].

Under the assumption that 10 ng of cfDNA contains approximately 3075 haploid genome copies (DNA content of one human cell: 6.5 pg), about 1540 individual cfDNA fragments are analyzed by our assay for each region of interest [28]. Considering a panel of 30 target regions, about 3–6 mutant reads can be detected in a plasma sample at a ctDNA concentration of 0.01%. In our experience, this is usually enough to detect treatment response to radiotherapy, even in localized tumors or small lung metastases with a volume of less than 1 cm^3^ (see Figure 1E, Figure 2B and Figure 3B) [16]. ctDNA levels below this threshold represent a significant technical challenge and often remains elusive if these reads represent MRD or only false positive events, which tend to occur at 1–2 reads/sample with this approach [16].

## 4. Materials and Methods

### 4.1. Study Population

Samples in this study were obtained from three patients. These included three tumors (one primary and two metastases) and 25 blood samples collected during treatment. Patients were treated at the interdisciplinary Comprehensive Cancer Center Freiburg (CCCF, Freiburg, Germany). Formalin-fixed paraffin-embedded (FFPE) tumor tissue, fresh-frozen tissue, plasma, and whole blood samples were available for analysis.

### 4.2. Blood Sampling

All blood samples were collected by puncture of the antecubital vein without tourniquet through a 20-gauge needle. The first 3 mL of blood was discarded. Each 9 mL of whole blood was collected in K_2_EDTA (1.6 mg EDTA/mL blood) tubes (Sarstedt AG & Co, Nümbrecht, Germany). Blood was processed within 2 h after blood withdrawal. Blood samples were double centrifuged for 15 min at 2500 g at 22 °C. Plasma aliquots were stored in cryotubes (FluidX) at −80 °C before use.

### 4.3. Isolation of DNA from Native and FFPE Tissue

DNA was extracted from tumor native tissue according to the manufacturer’s instructions using the DNeasy Blood and Tissue Kit (Qiagen, Hilden, Germany). DNA from formalin-fixed paraffin-embedded (FFPE) tissue was extracted using the FFPE Qiagen Extraction Kit. Approximately 8 sections 5–10 µm were digested with Proteinase K at 56° C for 3 days, and then eluted in 80 µL RNase-free water.

### 4.4. Isolation of DNA from Blood/Leukocytes

DNA was extracted with the DNeasy Blood and Tissue Kit (Qiagen, Hilden, Germany). The DNA was eluted in a volume of 200 µL RNase-free water.

### 4.5. Cell-Free DNA Isolation

Cell-free DNA was extracted from 1 mL to 5 mL of plasma using the QIAamp Circulating Nucleic Acid Kit (Qiagen, Hilden, Germany) according to the manufacturer’s protocol. An elution in AVE buffer, with a volume of 30 µL, was used. Purified cfDNA was stored in LoBind DNA tubes (Eppendorf, Hamburg, Germany) at −20 °C.

### 4.6. Quantification of cfDNA

The quantity of cfDNA was determined with a Qubit 3.0 Fluorometer using a high-sensitivity Qubit dsDNA HS Assay Kit (Invitrogen, Carlsbad, CA, USA).

### 4.7. Tumor Exome Sequencing

Tumor exome sequencing was performed at CeGaT (Tübingen, Germany). During sequencing library generation, samples were enriched using the Twist Human Core Exome Plus Kit (Twist Bioscience, San Francisco, USA). Subsequently, the samples were sequenced on an Illumina platform. A matched normal sample from whole blood DNA was sequenced as control.

### 4.8. Library Preparation for Tumor, Leukocyte, and cfDNA Samples and Next Generation Sequencing

Libraries from cfDNA were generated according to the manufacturer’s manual with a Takara SMARter Thruplex Tag Seq 48S Kit (Takara Bio In., Kusatsu, Shiga, Japan). A total of 10 ng of cfDNA input was used and adapters were ligated to attach unique molecular identifiers (UMI) to each side of the DNA fragment with single indices.

### 4.9. Panel Design and Hybridization

Tumor-specific target panels were designed based on the mutations identified in exome sequencing. Individual IDT xGen panels comprising biotin probes were used to hybridize target regions. DNA concentrations of the libraries were measured with Qubit (Invitrogen, Carlsbad, CA, USA) and equal amounts of each library were pooled (between 100–250 ng per library). Target regions were pulled down with streptavidin-coated magnetic beads. The hybridization protocol was performed according to the manufacturer’s instructions using an xGen Hybridization and Wash kit (Integrated DNA Technologies, Coralville, IA, USA). Following double hybridization-capture and PCR with 16 and 9 cycles, post capture samples were size selected using AMPure XP beads (Beckman Coulter, Brea, CA, USA). Capture was performed with libraries of 100–250 ng pooled to a total 500 ng–1 µg [25]. Both captures were incubated for 4 h at 65 °C.

### 4.10. Sequencing

The length of the cleaned up captured libraries was measured with a Tape Station Agilent D500 (Agilent, Santa Clara, CA, USA). For accurate DNA concentration, libraries were measured with a qPCR Light Cycler 480 System (Roche Diagnostics, Basel, Switzerland) using the NEBNext Library Quant Kit for Illumina (New England Biolabs, Ipswich, MA, USA).

The desired amount of the libraries was then calculated and sequenced using a MiSeq system with paired-end reads (MiSeq V2 300 cycle, Illumina Inc., San Diego, CA, USA).

### 4.11. Bioinformatics

Data analysis of exome sequencing was performed using a modified version of the MIRACUM pipe [29]. In brief: Quality control of the reads was performed using FASTQC [30]. Illumina adaptors, as well as trailing bases with low quality, were removed with a Trimmomatic [31]. The trimmed reads were aligned to the hg19 reference genome [32] using the BWA-MEM algorithm [33] and converted to a sorted BAM format using SAMtools [34]. Both normal tissue and tumor-specific variants were identified using Varscan [35] and annotated using Annovar [36]. For the rest of the procedure, only rare (global allele frequency < 0.1% based on GnomAD [37]) somatic mutations with at least four variant-specific reads and a variant allele frequency (VAF) > 10% were used. Subsequently, a patient-specific panel was generated for the genomic coordinates of the 30 somatic variants with the highest VAF.

The sequence reads of cfDNA samples containing UMIs were uploaded to the Curio Genomics web-platform (https://curiogenomics.com, accessed 5 May 2021) and aligned to GRCh38 using Bowtie2. UMI family consensus reads were called and somatic mutations including single nucleotide variants were discovered with the Curio Genomics built-in analysis pipeline. A minimum of two consensus reads were necessary for mutation calling [16]. In order to obtain general information about the performance of the panels, the sequence reads were further processed with the Illumina DRAGEN Enrichment Pipeline (BaseSpace, Illumina Inc., San Diego, CA, USA). All events were documented in the GRCh38 assembly.

### 4.12. Tumor Volume Rendering

Regions of interest (ROIs)-based data processing was performed with the in-house platform NORA (www.nora-imaging.com, accessed 5 May 2021). Primary tumors and metastases were segmented on consecutive slices on axial T2 weighted MRI sequences and on multidetector CT by one trained radiology resident. The tumors and metastases were delineated against normal tissue and volume renderings were calculated.

## 5. Conclusions

In this study, we present an approach for ctDNA monitoring in STS patients independent of the histological subtype for use in a routine diagnostic setting. Quantification of ctDNA on the basis of cancer exome profiling helps to predict tumor recurrence and monitor treatment response with minimal invasiveness at affordable cost. Given our promising results, the methods we describe warrant investigations in larger trials. The goal is to implement our methods in a clinical routine setting in the near future.

## Figures and Tables

**Figure 1 ijms-23-10215-f001:**
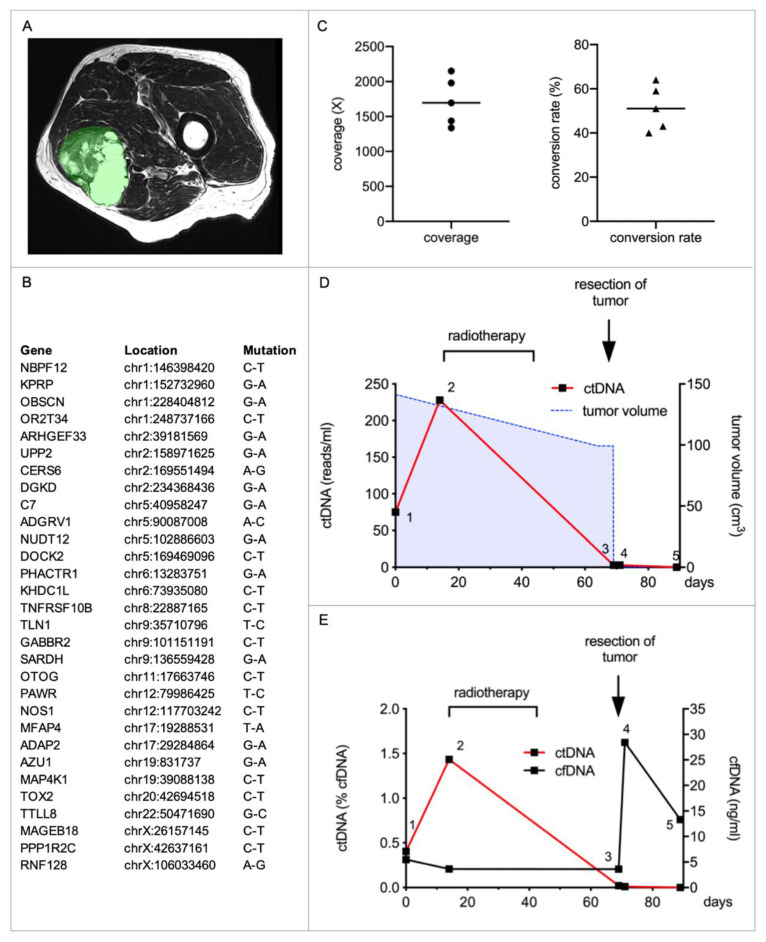
**ctDNA enables monitoring of the treatment response of neoadjuvant radiotherapy and tumor resection in a patient with myxofibrosarcoma.** (**A**) The patient presented with a primary tumor (tumor volume: 141.2 cm^3^) of his left thigh. Shown is a representative section of the MRI imaging with the tumor highlighted in green. After interdisciplinary case discussion, neoadjuvant radiotherapy was initiated before the tumor was surgically removed. (**B**) An individual mini-panel covering 30 mutations identified by exome sequencing was established for targeted NGS. Genomic coordinates are based on GRCh37 (hg19). (**C**) Five plasma samples were collected during his treatment, cfDNA was isolated, and ctDNA was therein quantified. Depicted are the coverage (circles) and conversion rates (triangles) of each sample after building UMI consensus families. Means are illustrated by horizontal bars. (**D**) Absolute quantification of ctDNA during the course of his treatment. Sample 1 and 2 were obtained before initiation of radiotherapy, sample 3 before surgery, and sample 4 and 5 after tumor removal. Depicted are absolute amounts of ctDNA (red line). The tumor volume, as calculated from MRI imaging, is depicted by the blue dashed line. (**E**) Depicted are the cfDNA concentrations (black line) and relative amounts of ctDNA (red line) in the same five plasma samples.

**Figure 2 ijms-23-10215-f002:**
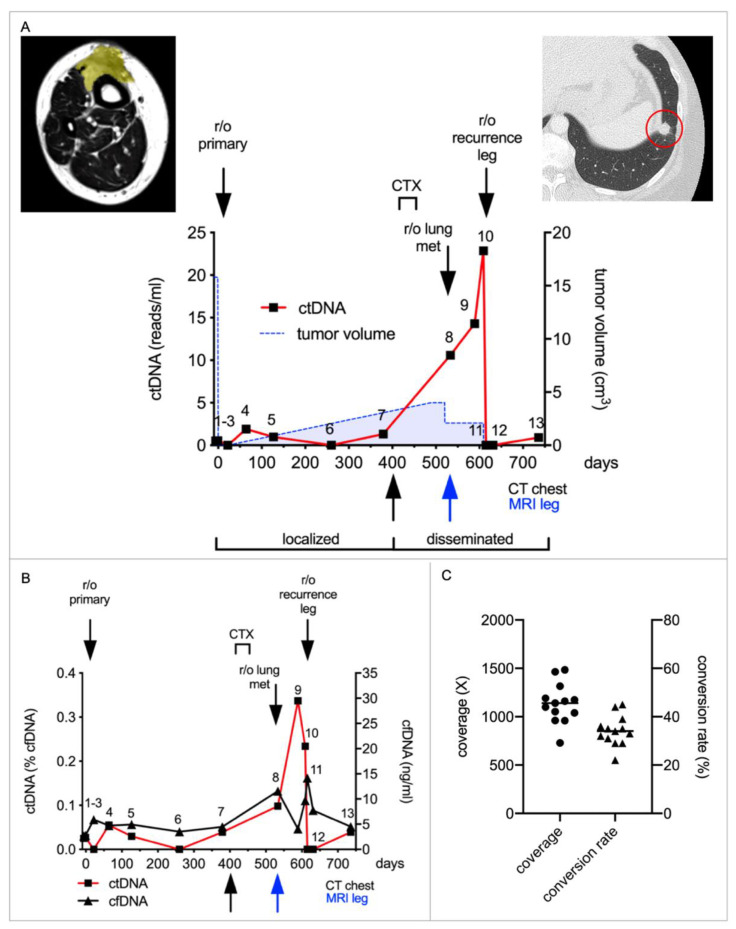
**ctDNA enables the detection of local and distant metastases in a patient with leiomyosarcoma.** (**A**) The patient presented with a primary tumor (yellow, volume: 15.8 cm^3^) on her shin which infiltrated the tibia as demonstrated in the MRI image (left). After neoadjuvant radiotherapy, the tumor was resected. Routine follow-up according to ESMO guidelines was conducted. In the second year after removal of the primary tumor, lung metastases and a local recurrence were detected. As there was progressive disease with chemotherapy (CTX), several lung metastases and the recurrence were resected. An individual mini-panel covering seven mutations identified by exome sequencing was established to quantify ctDNA in 13 plasma samples. Absolute quantification revealed low levels of ctDNA during follow-up, indicating MRD (sample 4–7). Levels rose steadily during progressive disease (sample 8–10) and again decreased after tumor resections (sample 11 and 12). Depicted are absolute amounts of ctDNA (red line). The tumor volume, as calculated from MRI and CT imaging, is depicted by the blue dashed line. Colored arrows indicate the timepoints of CT chest and MRI imaging, at which the recurrence/metastases were first detected. The red circle in the right subfigure demonstrates the representative metastasis in CT chest. (**B**) Depicted are the cfDNA concentrations (black line) and relative amounts of ctDNA (red line) in the same samples. (**C**) shows the coverage (circles) and conversion rate (triangles) of each sample after building UMI consensus families. Means are illustrated by horizontal bars.

**Figure 3 ijms-23-10215-f003:**
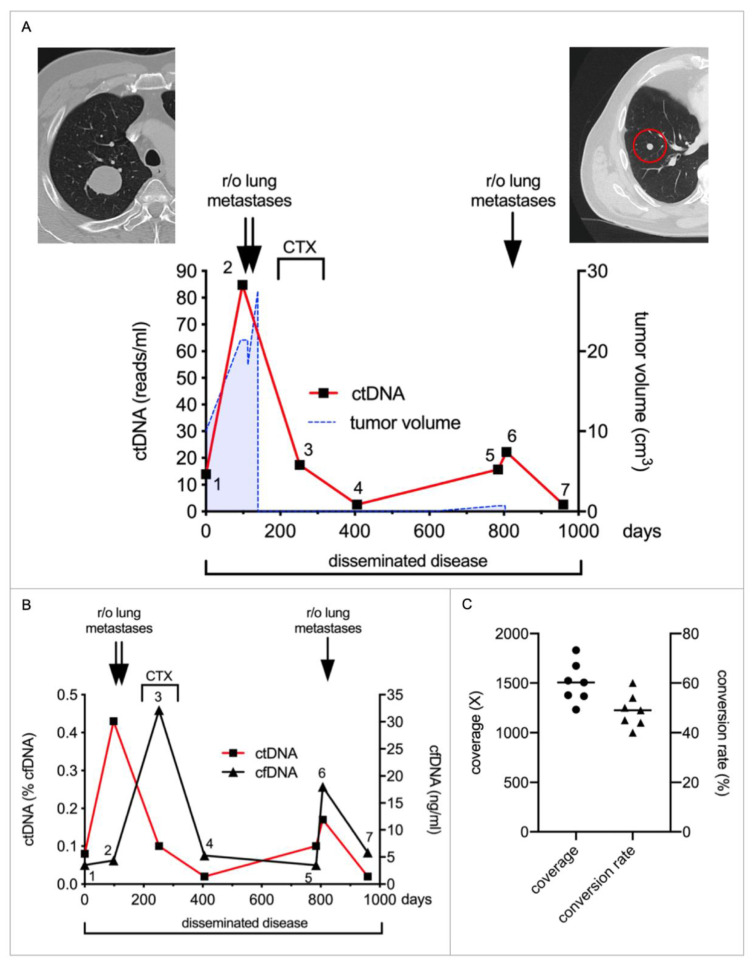
**Monitoring of treatment response in a patient with a metastatic Undifferentiated Pleomorphic Sarcoma.** (**A**) The patient suffered from an Undifferentiated Pleomorphic Sarcoma of the thigh, which was resected 1 year prior to inclusion in the study. On presentation, lung metastases were suspected on CT imaging (left upper image) and subsequently removed on two separate operations (double arrows). Treatment was followed by adjuvant chemotherapy (CTX). The patient was in complete remission until a new lung lesion was detected 1.5 years later (red circle in right upper CT image). This lesion was again resected (single arrow). We obtained seven plasma samples during the course of his treatment. ctDNA inclined with increasing tumor burden and declined after tumor resections and during CTX (red line). It never reached baseline following treatment, indicating sustained MRD. Tumor volume, as deduced from repeated imaging, is shown in blue. (**B**) Depicted are the cfDNA concentrations (black line) and relative amounts of ctDNA (red line) in the same samples. (**C**) shows the coverage (circles) and conversion rate (triangles) of each sample after building UMI consensus families. Means are illustrated by horizontal bars.

## Data Availability

The data presented in this study are available on request from the corresponding author. The data are not publicly available due to privacy restrictions.

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
