# Peer review of "Individualized Mini-Panel Sequencing of ctDNA Allows Tumor Monitoring in Complex Karyotype Sarcomas"

_ijms, 2022, doi:10.3390/ijms231810215_

Round 1
Reviewer 1 Report
The author presents a method the CTDNA in a rare tumor STS and suggested ctDNA has advantages as a predictable marker for treatment decisions. However, the assay was done on three patients with three different types of diseases and three different treatment plans. The only conclusion the author can drive is : the ctDNA is detectable in STS. All the other conclusions in the manuscript are not supported by the data. Espeically, since there is no healthy control here, how does the author know these CT/CFDNA are from the tumor but not from healthy tissues? If the DNA were released by non-tumorous tissue, how can we use it to predict treatment decisions? It may be only a marker for evaluating radiotherapy collateral damage. In addition, the author thinks that the patient with recurrence showed lower levels of ctDNAc compared to pretreatment, since there are no statistics, this is an unreliable conclusion.
Author Response
please see attached PDF file

Reviewer 2 Report
Although a small series, This shows how collaboration between a lab and clinicians can help understand clinical course of rare sarcomas. These results show cfDNA and ctDNA analysis using commercially available kits can be reliably done AND may help in the long term monitoring for detection of MRD in sarcomas. I would encourage this group to do the same in osteosarcoma to show general applicability of this thoughtful approach and submit another Paper to IJMS!
The only suggestion would be to expand background information and discussion about how some sarcomas have very complex karyotypes that result in many mutations being available for analysis in contrast to other sarcomas where gene fusion and a paucity of mutations is the predominant pattern. In the sarcomas with many this mutations provide an opportunity for validation of attempts to discern patterns of circulating tumor DNA at diagnosis, during therapy, and if sensitive and specific at relapse before this can be detected by conventional CT scans. THis study shows that this concept is valid.
Reviewer 3 Report
The authors present an approach for ctDNA monitoring in STS patients independent of the histological subtype for use in a routine diagnostic setting. Although the results in this study are promising, this manuscript is very interesting. My comments are as follow.
1) The authors described that quantification of ctDNA allows the early detection of recurrence or metastases and can be used to monitor treatment response in STS. Is ctDNA more useful for monitoring STS than cfDNA? If so, the title should be changed. The authors also should discuss the difference in results between ctDNA and cfDNA.
2) Figure 2 is missing in the manuscript.
3) Is the conversion rate between 20 – 60 % in this study reasonable? If so, please provide references to support this.
Round 2
Reviewer 1 Report
The author demonstrated that this is a proof of concept assay and provided some references and external databases to support their conclusions. Generally, this is an acceptable answer to most of my questions. The author has made a significant amount of revisions to the manuscript to make it fit the data better. Generally, the author has addressed most of my questions.